# Macrophage-like Cells Are Increased in Patients with Vision-Threatening Diabetic Retinopathy and Correlate with Macular Edema

**DOI:** 10.3390/diagnostics12112793

**Published:** 2022-11-15

**Authors:** Nigel T. Zhang, Peter L. Nesper, Janice X. Ong, Jacob M. Wang, Amani A. Fawzi, Jeremy A. Lavine

**Affiliations:** The Department of Ophthalmology, Feinberg School of Medicine, Northwestern University, 240 E. Huron Street, Bldg. McGaw M343, Chicago, IL 60611, USA

**Keywords:** diabetic retinopathy, diabetic macular edema, macrophage, macrophage-like cell, optical coherence tomography (OCT), OCT-angiography

## Abstract

Macrophage-like cells (MLCs) are potential inflammatory biomarkers. We previously showed that MLCs are increased in proliferative diabetic retinopathy (PDR) eyes. Vision-threatening diabetic retinopathy (VTDR) includes PDR, severe non-PDR (NPDR), and diabetic macular edema (DME). No prior data exist on MLCs in eyes with severe NPDR or DME. This prospective, cross-sectional optical coherence tomography-angiography (OCT-A) imaging study included 40 eyes of 37 participants who had NPDR classified as non-VTDR (*n* = 18) or VTDR (*n* = 22). Repeated OCT-A images were registered, averaged, and used to quantify the main outcome measures: MLC density and percent area. MLC density and percent area were correlated with clinical characteristics, NPDR stage, presence of DME, and OCT central subfield thickness (CST). In VTDR eyes, MLC density (2.6-fold, *p* < 0.001) and MLC percent area (2.5-fold, *p* < 0.01) were increased compared with non-VTDR eyes. Multiple linear regression analysis between MLC metrics and clinical characteristics found that MLC density was positively correlated with worse NPDR severity (*p* = 0.023) and higher CST values (*p* = 0.010), while MLC percent area was only positively associated with increased CST values (*p* = 0.006). MLCs are increased in patients with VTDR. Macular edema is the most strongly associated factor with increased MLC numbers in NPDR eyes.

## 1. Introduction

Diabetic macular edema (DME) is a significant cause of vision loss in the working age population. Anti-vascular endothelial growth factor (VEGF) is a safe and effective first-line therapy for DME [1]. However, 32–66% of DME patients demonstrate persistent macular edema after 6 monthly intravitreal anti-VEGF injections [2]. Persistent macular edema is hypothesized to be caused by untreated inflammation, which is supported by the effective role for steroids to treat DME [3] and remove persistent macular edema in anti-VEGF resistant patients [4]. In support of this concept, reduced classical monocyte recruitment to the retina reduces diabetic retinopathy (DR) progression [5], while *Cx3cr1*-deficiency, which impacts microglia homeostasis, accelerates DR progression in mice [6]. These findings suggest that inflammation, and specifically macrophages, play a key role in DR pathogenesis and that inflammatory biomarkers are an important step toward personalized medicine.

Macrophage-like cells (MLCs) are detectable at the vitreoretinal interface using clinical optical coherence tomography angiography (OCT-A) [7] and adaptive optics [8] imaging. Using repeated and averaged OCT-A, a thin three-micron structural OCT slab above the internal limiting membrane (ILM) can reliably image ramified MLCs on the surface of the retina because of the dark contrast of the vitreous. This is not possible within the neural retina because of the lack of contrast. OCT-A is important for this analysis because the vascular plexus is used for registration of multiple scans. MLCs possess properties associated with macrophages including mobility, a ramified morphology [9], and correspondence to *Cx3cr1*^+^ cells (a macrophage marker) in mice [10]. Furthermore, we recently showed that clinical MLCs correspond to microglia, vitreal hyalocytes, and perivascular macrophages at steady state, and additionally include classical monocytes and monocyte-derived macrophages during neuroinflammation [11]. Furthermore, MLCs are increased in eyes with Behcet’s uveitis and are correlated with worsened fluorescein leakage, suggesting their role in retinal inflammation [12]. Clinically, we previously showed that MLC density is increased in eyes with proliferative DR (PDR) in comparison with healthy eyes, eyes with diabetes but no DR, and mild-moderate non-PDR (NPDR) eyes [13]. However, our prior study lacked eyes with severe NPDR and DME. In addition, MLCs are up-regulated in eyes with retinal vein occlusion (RVO) and correlate with both macular edema and ischemia [14]. Based upon these data, we hypothesized that MLC numbers would be increased in vision-threatening diabetic retinopathy (VTDR), which includes severe NPDR and mild-moderate NPDR with DME. VTDR is a meaningful outcome with high risk of short-term vision loss [15].

We performed repeated OCT-A imaging on NPDR patients, classified as with and without VTDR. We hypothesized that MLCs would be increased in VTDR eyes. Next, we correlated clinical, demographic, ocular, and imaging characteristics with MLC metrics to determine which factors are most associated with MLCs using both univariate and multivariate analyses.

## 2. Methods

This prospective, cross-sectional study enrolled patients with NPDR seen between October 2019 and February 2022 in the Department of Ophthalmology at Northwestern University in Chicago, Illinois. Written informed consent was obtained from all subjects before participation. The study was approved by the Institutional Review Board of Northwestern University (IRB no. STU00200890) and conducted in accordance with the tenets of the Declaration of Helsinki and the regulations of the Health Insurance Portability and Accountability Act.

Inclusion criteria were patients 18 years of age or greater, with any stage of NPDR. NPDR eyes were staged by chart review and verified by JAL according to the International Clinical Diabetic Retinopathy Disease Severity Scale, using color fundus photographs and/or fluorescein angiography obtained using the Ultra-Widefield Scanning Laser Ophthalmoscope (Optomap Panoramic 200; Optos PLC, Scotland, UK). Exclusion criteria were vitreomacular pathology like epiretinal membrane or vitreomacular traction, other retinal disease such as age-related macular degeneration, or significant media or lens opacities that would prevent high-quality imaging. From the review of the electronic health record, we recorded age, sex, hemoglobin A1c, diabetes duration, diabetes type, NPDR stage, BCVA, number of prior intravitreal injections, and presence of DME. DME presence was defined as a history of intravitreal injections for DME or current CST >320 microns in men or >305 microns in women on Spectralis OCT [16] (Heidelberg Engineering Inc., Heidelberg, Germany).

### 2.1. OCT-A Imaging

Patients underwent repeated (mean: 5.8, range: 3–9) spectral domain (SD)-OCTA images over a nominal scan area of 3 mm × 3 mm (304 pixels × 304 pixels) centered on the fovea using the RTVue-XR Avanti system (Optovue, Inc., Fremont, CA, USA) with split-spectrum amplitude-decorrelation angiography (SSADA) software (version 2017.1.0.151), as previously described [13,17]. Images with motion artifact or a Q score less than 5 were excluded. For subjects with imaging in both eyes, both eyes could be enrolled in the study if their NPDR stage or DME presence was disparate. If NPDR stage and DME presence were identical, the eye with higher image quality (average Q score) was included.

### 2.2. MLC Quantification

Image registration was performed using the full retinal vascular network OCT-A slab via the Register Virtual Stack Slices Plugin (Feature Extraction Model = Rigid, Registration Model = Elastic) in FIJI, a distribution of the program ImageJ (National Institutes of Health, Bethesda, MD, USA). We segmented the slab from 0 to 3 microns above the ILM to detect MLCs, as previously described [7]. The saved transformation matrix was applied to the MLC layer slabs using the Transform Virtual Stack Slices Plugin. Registered MLC stacks were then averaged using the Z-project plugin. MLCs were identified from averaged images using our previously published [13] semiautomated custom macro in FIJI. This approach uses noise reduction to remove background irregularities and vessel artifacts, followed by signal enhancement for improved MLC identification, and finally binarization to extract discrete cell shapes [13]. We quantified the MLC density and percent area using the Analyze Particles function. MLC density was defined as the number of cells per mm^2^ and MLC percent area was defined as the percentage of the scan comprised of MLCs. We investigated both metrics because it is difficult to reliably detect the boundary between two adjacent cells when each pixel is approximately 10 microns. MLC quantification was performed by two graders (N.T.Z. and P.L.N.) who were masked to NPDR stage and DME history.

### 2.3. Superficial Vascular Plexus Perifoveal Vessel Length Density

We calculated perifoveal vessel length density (VLD) of the superficial vascular plexus (SVP), similarly to as previously described [18]. The averaged SVP OCT-A slab was binarized using mean thresholding and skeletonized. A 1 mm diameter circle was centered on the fovea to delete the foveal avascular zone and parafoveal area. The remaining skeletonized vascular map area was measured and expressed as a ratio compared with the total measured image area.

### 2.4. Statistics

Statistical analysis was performed using GraphPad Prism 9.0.1 (GraphPad Software, San Diego, CA, USA). Dataset normality was determined using the Shapiro–Wilk test. MLC density and percent area were both normally distributed. Demographic information was compared using Student’s unpaired *t*-test for continuous variables or Chi-square test for categorical variables. Pearson correlation was used for association between current (NTZ) and prior (PLN) data from Ong et al. [13]. Student’s unpaired *t*-test was used for comparison of MLC density and percent area between VTDR and non-VTDR eyes. Pearson correlation was used for univariate analysis between MLC parameters and continuous variables: age, hemoglobin A1c, diabetes duration, CST, BCVA, number of prior intravitreal injections, and SVP perifoveal VLD. Spearman correlations were used for univariate analyses between MLC metrics and categorical variables: sex, diabetes type, NPDR stage, and DME presence. Multiple linear regression analysis with the least squares method was used for multivariate analysis. We included any variable with a *p*-value < 0.05 (vs. MLC density or MLC percent area) in the univariate analysis, including age, diabetes type, NPDR stage, presence of DME, and CST. One-way ANOVA followed by Tukey’s multiple comparisons test was used to compare MLC counts and NPDR stage.

## 3. Results

This prospective, cross-sectional OCT-A imaging study included 40 eyes of 37 participants who had NPDR classified as non-VTDR (*n* = 18) or VTDR (*n* = 22). The VTDR group included eyes with mild NPDR with DME, moderate NPDR with DME, and severe NPDR with or without DME. The non-VTDR group included 12 mild NPDR without DME and 6 moderate NPDR without DME eyes (Table 1). The VTDR group was comprised of three mild NPDR with DME, nine moderate NPDR with DME, six severe NPDR eyes without DME, and four severe NPDR with DME eyes. The groups were well matched in terms of sex, refractive error, and LogMAR BCVA (Table 1). On average, the VTDR group was 10 years older, included more patients with Type 2 diabetes, showed longer diabetes duration, demonstrated worse hemoglobin A1c, had thicker average CST, and had more intravitreal injections compared with the non-VTDR group (Table 1). Each of these group imbalances, except diabetes type, were expected in eyes with VTDR compared with non-VTDR eyes.

Representative non-VTDR (A-C) and VTDR (D-F) eyes are shown in Figure 1. We found that VTDR eyes demonstrated 2.6-fold (*p* < 0.001) increased MLC density and 2.5-fold (*p* < 0.01) greater MLC percent area compared with non-VTDR eyes (Figure 1A–H). To verify the inter-rater reliability, 18 eyes were analyzed independently by NTZ (*y*-axis) and PLN (*x*-axis, Ong et al. [13]). MLC densities were strongly correlated between graders (r = 0.878, *p* < 0.001, Figure 1I), confirming the high reproducibility of our methodology.

As VTDR eyes were different from non-VTDR eyes in terms of age, diabetes type, diabetes duration, diabetes control, NPDR stage, presence of DME, and number of prior intravitreal injections (Table 1), we conducted a univariate analysis between MLC parameters and clinical characteristics to determine the features associated with increased MLCs in VTDR eyes. We performed correlation analysis between MLC density, MLC percent area, age, sex, hemoglobin A1c, diabetes duration, diabetes type, NPDR stage, DME presence, CST, BCVA, number of prior intravitreal injections, and SVP perifoveal VLD. As MLCs exist on the surface of the retina, we analyzed the superficial vascular plexus because of its close anatomical association. We chose to analyze skeletonized VLD to equally count large vessels and smaller capillaries [18,19]. We found that MLC density and MLC percent area were highly correlated (r = 0.930, *p* < 0.001, Table 2). Neither MLC parameter was significantly correlated with sex, hemoglobin A1c, diabetes duration, BCVA, number of intravitreal injections, or SVP VLD (Table 2). MLC density was positively associated with increased age (r = 0.333, *p* = 0.036), type 2 diabetes (r = 0.527, *p* < 0.001), more severe NPDR stage (r = 0.331, *p* = 0.037), and higher CST values (r = 0.414, *p* = 0.008). MLC percent area was also positively correlated with older age (r = 0.340, *p* = 0.032), type 2 diabetes (r = 0.499, *p* = 0.001), DME presence (r = 0.332, *p* = 0.037), and thicker CST values (r = 0.483, *p* = 0.002).

To determine which of these variables most influenced MLC density and percent area, we conducted a multiple linear regression analysis between MLC density or MLC percent area and age, diabetes type, NPDR stage, DME presence, and CST (Table 2). MLC density remained positively correlated with worse NPDR severity (*p* = 0.023) and higher CST values (*p* = 0.010). MLC percent area was only positively associated with increased CST values (*p* = 0.006).

To confirm the effect of NPDR stage upon MLC parameters, we then grouped patients by NPDR stage alone. Representative mild (A–C), moderate (D–F), and severe (G–I) NPDR eyes are shown in Figure 2. Average MLC density and percent area increased with more severe NPDR stage (Figure 2A–I). However, MLC density was only significantly different between mild and severe NPDR eyes (Figure 2J). MLC percent area was not significantly different between groups (Figure 2K), in agreement with the results of the multivariate analysis.

To confirm the influence of macular edema on MLC density and percent area, we plotted MLC parameters versus CST for all patients. Representative eyes with thin (A) and thick (B) CST are shown in Figure 3. The number of MLCs was greater in eyes with thicker compared with thinner CST (Figure 3A,B). MLC density (r = 0.414, *p* = 0.008, Figure 3C) and percent area (r = 0.483, *p* = 0.002, Figure 3D) both showed a significant positive correlation with CST. These data suggest that MLC metrics correlate with macular edema severity.

## 4. Discussion

DR and DME are vascular diseases with a significant inflammatory component [5,6]. We previously showed that MLC density was increased in patients with PDR [13]. However, our prior study lacked patients with DME and severe NPDR. We thus investigated whether eyes with VTDR, including severe NPDR, mild NPDR with DME, and moderate NPDR with DME, had increased MLC parameters. We found that MLC density and percent area were increased 2.6- and 2.5-fold, respectively, in eyes with VTDR compared with non-VTDR eyes (Figure 1). Multivariate analysis found that NPDR stage was correlated with MLC density, and higher CST values were positively associated with MLC density and percent area (Table 2, Figure 2 and Figure 3). Therefore, DME is a key associated factor with increased MLC density and percent area in NPDR eyes with VTDR.

We did not find an association between MLC metrics and SVP perifoveal VLD (Table 2); however, ischemia could still be an important MLC driver. We previously showed that MLC density was increased in PDR eyes with an average density of 21.9 cells per mm^2^ [13]. In this study, VTDR eyes displayed an MLC density of 12.0 cells per mm^2^, which is less than PDR eyes. In VTDR eyes, SVP perifoveal VLD, which correlates with geometric perfusion deficits and ischemia, did not influence MLCs (Table 2). However, given our prior results of very high MLC counts in PDR eyes without DME, ischemia is still a likely important contributor to increased MLCs in PDR eyes. Future investigations into the impact of ischemia metrics in different vascular plexuses and areas outside the macula are necessary to further identify the role of ischemia in MLC numbers.

The non-VTDR group was skewed toward patients with Type 1 diabetes. Interestingly, the non-VTDR group had a longer diabetes duration and better diabetes control compared with the VTDR patients (Table 1). These data suggest that the patients with Type 1 diabetes with mild–moderate NPDR without DME were extremely well controlled, preventing DR progression. However, diabetes type, duration, and control were not associated with MLC density or percent area upon multivariate analysis.

We recently investigated the identity of MLCs in mice. During steady state, MLCs include microglia, perivascular macrophages, and vitreal hyalocytes [11]. After monocyte chemoattract protein-1 (MCP-1)-driven neuroinflammation, which promotes monocyte infiltration into tissue, MLCs also include classical monocytes and monocyte-derived macrophages [11]. MCP-1 is consistently increased in aqueous and vitreous samples from patients with DR, and MCP-1 correlates with macular edema [20]. Furthermore, patients who are poor responders to anti-VEGF medications demonstrate low levels of VEGF and high levels of MCP-1 [21]. Additionally, monocyte-derived macrophages promote DR progression in mice [5]. These data suggest that increased MLCs in VTDR eyes could include pathogenic monocyte-derived macrophages that promote inflammation and drive resistance to anti-VEGF medications during DME. In support of this hypothesis, MLCs are increased in eyes with Behcet’s uveitis and correlated with more fluorescein leakage, confirming that MLCs are increased in eyes with inflammation and without retinal vascular disease [12]. Thus, MLCs could be a biomarker of inflammation in eyes with DME.

Vitreal hyalocytes are another potential MLC identity. Hyalocytes are resident macrophages of the vitreous that present antigens, participate in immune privilege, and express pro-angiogenic and pro-chemotactic transcriptional profiles [22,23]. Thus, MLCs could include vitreal hyalocytes promoting angiogenesis and immune cell infiltration in DME eyes. Similar to monocyte-derived macrophages, we would hypothesize that vitreal hyalocytes are also pathogenic and a biomarker of inflammation.

However, MLCs could also be microglia. Microglia are self-renewing, tissue resident retinal macrophages that primarily exist in the inner and outer plexiform layers, but microglia are also present in the retinal nerve fiber layer [24,25]. Microglia dysfunction via *Cx3cr1*-deficiency worsens DR progression in mice [6]. Recent adaptive optics studies show that many MLCs are highly ramified and resemble microglia morphology [9]. Thus, increased MLCs in VTDR eyes could be microglia playing a protective or restorative role. Our data demonstrate that MLCs are correlated with DME, but do not demonstrate causality. Future studies are needed to determine the identity of MLCs during diabetes.

Finally, anotther identity includes perivascular macrophages, which function in blood–retinal barrier maintenance, vascular permeability, immune cell infiltration, and clearance of debris from the perivascular space [26,27]. The role of perivascular macrophages in DR and DME is unknown. MLCs in DME eyes could be perivascular macrophages regulating vascular permeability and/or immune cell infiltration in either pro- or anti-inflammatory directions. Future studies in animal models are necessary to unravel the role of each of these macrophage subtypes during DR to determine how MLC imaging informs clinicians about DR and DME pathogenesis.

Limitations of this study include its cross-sectional design and a lack of information regarding response to therapy. An area of ongoing investigation is how MLCs respond to DME treatments. Although it would be ideal to include these data, longitudinal follow-up of 6–12 months would be necessary, which is beyond the scope of this study. This important next step will help determine the function of MLCs during DME and if they are an inflammatory biomarker.

## 5. Conclusions

MLCs are increased in VTDR eyes and correlate with macular edema. Future studies are necessary to determine the role of MLCs in NPDR progression and DME pathogenesis.

## Figures and Tables

**Figure 1 diagnostics-12-02793-f001:**
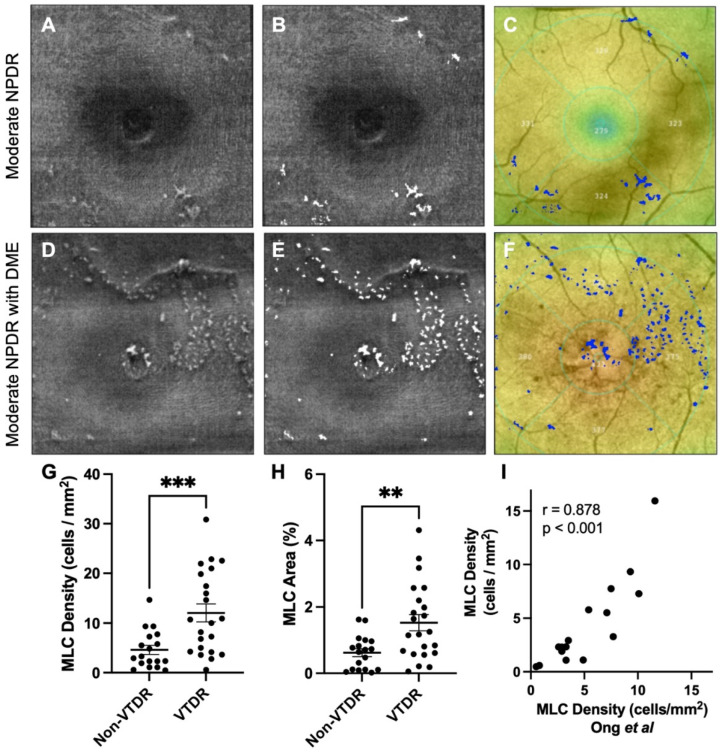
**VTDR eyes demonstrate increased MLC numbers**. First column from the left (**A**,**D**) shows MLC slab 0–3 microns above the internal limiting membrane. Second column (**B**,**E**) depicts quantitated MLCs (in white) overlayed on ILM scan. Third column (**C**,**F**) displays MLCs (in blue) overlayed on the macular thickness map. Each row demonstrates a representative moderate NPDR eye without DME (**A**–**C**) and moderate NPDR eye with DME (**D**–**F**). Eyes with VTDR showed increased MLC density (**G**) and area (**H**) compared with non-VTDR eyes. MLC density was strongly correlated between our current (*y*-axis) and prior (*x*-axis, Ong et al.) manuscripts (**I**). ** *p* < 0.01. *** *p* < 0.001.

**Figure 2 diagnostics-12-02793-f002:**
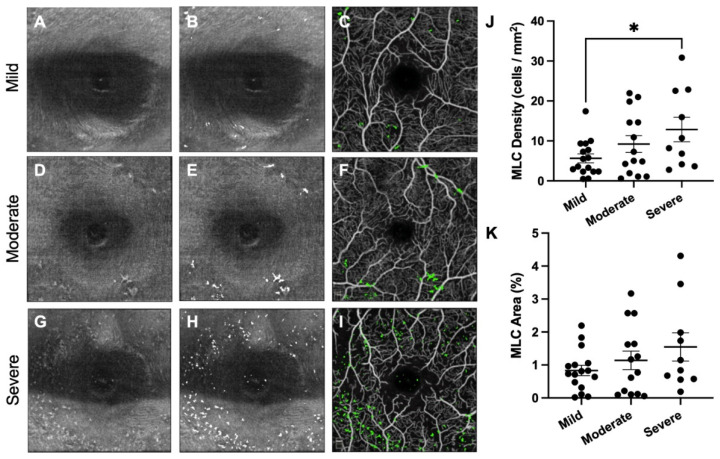
**MLC density is increased in severe NPDR eyes**. First column from the left (**A**,**D**,**G**) shows MLC slab 0–3 microns above the internal limiting membrane. Second column (**B**,**E**,**H**) depicts quantitated MLCs (in white) overlayed on ILM scan. Third column (**C**,**F**,**I**) displays MLCs (in green) overlayed on the SVP slab. Each row demonstrates a representative eye from the mild (**A**–**C**), moderate (**D**–**F**), and severe (**G**–**I**) NPDR groups. MLC density (**J**) was increased in severe NPDR eyes compared with mild NPDR eyes (* *p* < 0.05). MLC percent area (**K**) showed a trend toward increased area with worsened severity.

**Figure 3 diagnostics-12-02793-f003:**
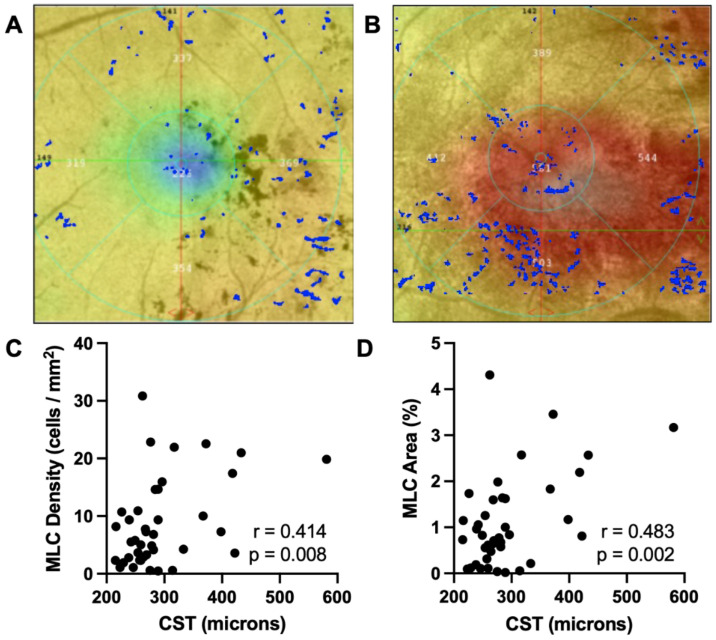
**MLCs are correlated with DME severity**. Representative overlays of MLCs (blue) on the macular thickness map (from thick to thin: white → red → yellow → green → blue) in a representative eye with low (**A**) and high CST (**B**). MLC density (**C**) and percent area (**D**) were strongly associated with CST.

**Table 1 diagnostics-12-02793-t001:** Demographic characteristics.

	Groups		*p*-Value
	Non-Vision Threatening Diabetic Retinopathy	Vision Threatening Diabetic Retinopathy	
**Number of Eyes**	18	22	
**NPDR Stage**	Mild 12 (67%) Moderate 6 (33%) Severe (0%)	Mild 3 (14%) Moderate 9 (41%) Severe 10 (45%)	
**DME**	0 (0%)	16 (73%)	
**Age (mean ± SD)**	49.5 ± 15.4	59.2 ± 12.8	0.041
**Sex, *n* Female (%)**	9 (50%)	10 (45%)	0.775
**Refractive Error (D mean ± SD)**	−2.20 ± 2.21	−1.39 ± 3.22	0.413
**Missing, *n* (%)**	5 (28%)	2 (9%)	
**DM Type, *n* Type 1 (%)**	12 (67%)	2 (9%)	0.014
**DM Duration (mean ± SD)**	21.9 ± 11.6	14.5 ± 10.3	0.046
**HbA1c (mean ± SD)**	7.1 ± 0.7	8.1 ± 3.0	0.043
**LogMAR BCVA**	0.10 ± 0.10	0.11 ± 0.14	0.242
**CST**	258 ± 23	322 ± 88	0.003
**Number of IVI**	0 ± 0	7.6 ± 12.5	0.017

All statistical tests were Student’s unpaired *t*-test except for sex and DM type, which were analyzed using the Chi-square test. D = diopter, DM = diabetes mellitus, DME = diabetic macular edema, HbA1c = hemoglobin A1c, NPDR = non-proliferative diabetic retinopathy, SD = standard deviation, BCVA = best-corrected visual acuity, CST = central subfield thickness, IVI = intravitreal injection.

**Table 2 diagnostics-12-02793-t002:** Correlation between MLC and patient parameters.

	MLC Density (Univariate)	MLC Density (Multivariate)	MLC Percent Area (Univariate)	MLC Percent Area (Multivariate)
**MLC Percent Area**	0.930 (*p* < 0.001)	--	0.930 (*p* < 0.001)	--
**Age**	0.333 (*p* = 0.036)	*p* = 0.848	0.340 (*p* = 0.032)	*p* = 0.997
**Sex ***	0.037 (*p* = 0.821)	--	0.037 (*p* = 0.821)	--
**HbA1c**	0.126 (*p* = 0.437)	--	0.083 (*p* = 0.613)	--
**DM Duration**	−0.151 (*p* = 0.359)	--	−0.120 (*p* = 0.468)	--
**DM Type ***	0.527 (*p* < 0.001)	*p* = 0.429	0.499 (*p* = 0.001)	*p* = 0.352
**NPDR Stage ***	0.331 (*p* = 0.037)	*p = 0.023*	0.203 (*p* = 0.209)	*p* = 0.115
**DME Presence ***	0.318 (*p* = 0.045)	*p* = 0.214	0.332 (*p* = 0.037)	*p* = 0.300
**CST**	0.414 (*p* = 0.008)	*p = 0.010*	0.483 (*p* = 0.002)	*p = 0.006*
**LogMAR BCVA**	0.064 (*p* = 0.731)	--	0.150 (*p* = 0.422)	--
**Number of IVI**	0.171 (*p* = 0.303)	--	0.133 (*p* = 0.427)	--
**SVP VLD**	−0.171 (*p* = 0.290)	--	−0.168 (*p* = 0.301)	--

For univariate analysis, the Pearson or Spearman r value is reported with *p*-value in parentheses. * Sex, DM Type, NPDR stage, and DME presence were correlated with Spearman coefficients. For multivariate linear regression, *p*-value is reported. MLC = macrophage-like cell, SVP = superficial vascular plexus, VLD = vessel length density, DM = diabetes mellitus, DME = diabetic macular edema, HbA1c = hemoglobin A1c, NPDR = non-proliferative diabetic retinopathy, CMT = central macular thickness, BCVA = best-corrected visual acuity, IVI = intravitreal injection.

## Data Availability

The datasets used and analyzed are available from the corresponding author upon reasonable request.

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
