# Peer review of "Macrophage-like Cells Are Increased in Patients with Vision-Threatening Diabetic Retinopathy and Correlate with Macular Edema"

_diagnostics, 2022, doi:10.3390/diagnostics12112793_

Round 1

Reviewer 1 Report

The authors produced a very well written manuscript describing the quantification of macrophage like cells (MLCs) at the vitreous - inner retina border using OCTA (taking advantage of the temporal dynamics of these cells) in patients with varying degrees of non-vision threatening diabetic retinopathy and vision threatening diabetic retinopathy.

Although the results are are not surprising given their previous findings in Ong, J. X. et al. IOVS (2021), it is important to quantify these more severe cases of non-proliferative diabetic retinopathy and diabetic retinopathic patients that suffered from edema due to treatment resistance. 

I would have liked to have seen in the discussion why you think there is a larger number / percentage of subjects with type-1 diabetes that have non-vison threatening cases of DR than vision threatening cases? It seems that patients with type-2 diabetes are at higher risk to develop vision threatening DR. My guess would be that by the time you develop type-2 diabetes the damage is already done, while with type-1, patients are conscientious about their diet from the time of disease onset making them less likely to develop non-VTDR. 

I only found one typo in section 2.3. "binarizing" should be changed to "binarized".

Overall, the paper was clear and well written, and the study was well conducted. I accept this manuscript for publication

Author Response

Reviewer 1:

The authors produced a very well written manuscript describing the quantification of macrophage like cells (MLCs) at the vitreous - inner retina border using OCTA (taking advantage of the temporal dynamics of these cells) in patients with varying degrees of non-vision threatening diabetic retinopathy and vision threatening diabetic retinopathy. Although the results are are not surprising given their previous findings in Ong, J. X. et al. IOVS (2021), it is important to quantify these more severe cases of non-proliferative diabetic retinopathy and diabetic retinopathic patients that suffered from edema due to treatment resistance. 

Response: Thank you to the reviewer for their comments and thorough review.

I would have liked to have seen in the discussion why you think there is a larger number / percentage of subjects with type-1 diabetes that have non-vison threatening cases of DR than vision threatening cases? It seems that patients with type-2 diabetes are at higher risk to develop vision threatening DR. My guess would be that by the time you develop type-2 diabetes the damage is already done, while with type-1, patients are conscientious about their diet from the time of disease onset making them less likely to develop non-VTDR. 

Response: Thank you to the reviewer for noticing this aspect of our data. A discussion section on this topic was added as the third discussion paragraph.

I only found one typo in section 2.3. "binarizing" should be changed to "binarized".

Response: This typo has been corrected, thank you for catching it.

Overall, the paper was clear and well written, and the study was well conducted. I accept this manuscript for publication

Response: Thank you to the reviewer for their comments and thorough review.

Reviewer 2 Report

In this paper, Zhang et al, highlighted an increase of macrophage-like cells in patients with VTDR and, more interestingly, they observed a correlation of the levels of these cells with macular edema.

Here, are my main concerns:

- Although these results are promising, since they can provide a biomarker for the early detection of the disease, in my opinion, this point is not clearly discussed. To clarify, a more detailed and well-structured focus on inflammation, together with a deeper comparison with the literature, should be included in the discussion section, after a proper re-organization of the whole paragraph.

- Although figures presented in the results are very interesting, I cannot find an adequate description in this section. Please, add.

- Do you think that the retrieved increase in MLC can be evaluated also using other kind of analysis (e.g. blood sampling, or similar)? In my opinion, this can constitute a key point in view of personalized medicine.

- In Materials and Methods, Section 2.4 (Statistics), page 3, line 140: Change Spearman correlated with Spearman correlations.

Author Response

Reviewer 2:

In this paper, Zhang et al, highlighted an increase of macrophage-like cells in patients with VTDR and, more interestingly, they observed a correlation of the levels of these cells with macular edema.

Here, are my main concerns:

- Although these results are promising, since they can provide a biomarker for the early detection of the disease, in my opinion, this point is not clearly discussed. To clarify, a more detailed and well-structured focus on inflammation, together with a deeper comparison with the literature, should be included in the discussion section, after a proper re-organization of the whole paragraph

Response: We reorganized the discussion section to progress through the cell types based upon their role. We previously organized this section as monocyte-derived macrophages, microglia, perivascular macrophages, and hyalocytes because that is the order of their abundance during neuroinflammation. However, it makes more sense as this reviewer points out to organize this section based upon function. It now progress from monocyte-derived macrophages (pathogenic) to hyalocytes (pathogenic) to microglia (restorative) to perivascular macrophages (unknown). We thank the reviewer for their assistance in improving the discussion.

- Although figures presented in the results are very interesting, I cannot find an adequate description in this section. Please, add.

Response: A formatting error caused the Table legends to be integrated into the results text. This has been corrected and additional detail on each figure has been added to the text. Thank you to the reviewer for detecting this error.

- Do you think that the retrieved increase in MLC can be evaluated also using other kind of analysis (e.g. blood sampling, or similar)? In my opinion, this can constitute a key point in view of personalized medicine.

Response: This is an excellent idea and we are planning to pursue it as the research progresses. We hypothesize that MLCs in DR are blood monocyte-derived macrophages and we may be able to detect changes in the blood monocytes of patients with VTDR. However, there are many experiments to perform before we get to this study. Specifically, we need to show that MLCs are monocyte-derived macrophages in animal models. We thank the reviewer for recognizing the future potential of our work.

- In Materials and Methods, Section 2.4 (Statistics), page 3, line 140: Change Spearman correlated with Spearman correlations.

Response: This typo has been corrected, thank you for catching it.

Reviewer 3 Report

The author tried to observe MLCs changes in diabetic retinopathy patients through OCTA, which could be useful for managing DM patients in clinical work. Here are my comments:

1.     In line 30, diabetic macular edema is one of the main symptoms in diabetic retinopathy changes. What does the authors mainly tried to emphasize by separate macular edema here?

2.     Line 64-Lin73 should be showed in METHODS and RESULTS, instead of Introduction section.

3.     In table 1, why the author made a row named “DME”, as they already stated that DME was VTDR only.

4.     The author should clarify the exact definition of VTDR and Non-VTDR, even though it had been explained simply in line 60-63.

5.     According to the results showed in Figure 2, it seems that MLCs more likely to  concentrated along with vessel arcade. Despite of DME, are there any differences only in macular area among mild-, moderate- and severe-NPDR? Did MLC would potentially be a useful indicator in raising clinical attention in DM patients?

6.     Refrain to repeat results in Discussion (In line 264-272). The author needs to emphases the meaning of present study.

7.     In Line 296-311, the author mentioned that increased MLCs would play as a protective way and may arise inflammations. According to the results showed in Figure 2, the severe the NPDR is, the more MLCs detected. The author may need to re-organize this paragraph (line 296-311) clearer.

8.     The author needs to add a Conclusion section in summarizing the key-findings or meaning of current study.

9.     The author need to address clearer the value of current research in indicating the management of DM patients.

Author Response

Reviewer 3:

The author tried to observe MLCs changes in diabetic retinopathy patients through OCTA, which could be useful for managing DM patients in clinical work. Here are my comments:

In line 30, diabetic macular edema is one of the main symptoms in diabetic retinopathy changes. What does the authors mainly tried to emphasize by separate macular edema here?

Response: We changed this sentence to simply say that DME is a significant cause of vision loss. DR can also cause vision loss through vitreous hemorrhage, tractional retinal detachment, and macular ischemia. However, since DME is the focus of the manuscript, we simplified the first sentence to focus upon DME. Thank you to the reviewer for clarifying this vague start to the manuscript.

Line 64-Lin73 should be showed in METHODS and RESULTS, instead of Introduction section.

Response: It is common to summarize results at the end of the introduction. However, since we have a summary at the start of the discussion, as this reviewer points out, we changed this section to describe the hypotheses tested by this study. Thank you for improving the flow of this manuscript.

In table 1, why the author made a row named “DME”, as they already stated that DME was VTDR only.

Response: We recognize that this might be a little redundant, but included this variable for full transparency of the data. Severe NPDR eyes without DME are inlucded in the VTDR group. We included DME as an independent row because 16 / 22 eyes in the VTDR group had DME.

The author should clarify the exact definition of VTDR and Non-VTDR, even though it had been explained simply in line 60-63

Response: We added the definition of VTDR again to the second sentence of the results section. Thank you to the reviewer for improving the clarity of the study.

According to the results showed in Figure 2, it seems that MLCs more likely to  concentrated along with vessel arcade. Despite of DME, are there any differences only in macular area among mild-, moderate- and severe-NPDR? Did MLC would potentially be a useful indicator in raising clinical attention in DM patients?

Response: In Fig 2, MLC density was increased in severe NPDR eyes compared to mild NPDR eyes. This difference was statistically significant. However, MLC area was not significantly different, although a trend did exist. In comparison, CST was significantly correlated with MLC density and MLC percent area. So the data in this study suggests that CST is a more significant association than NPDR stage. We are currently conducting a larger study on a wide range of DR eyes without DME to investigate the effects of ischemia on MLC more thoroughly. But this is not an emphasis of this current manuscript. We emphasized the correlation between MLC density and NPDR stage in the first paragraph of the discussion.

Refrain to repeat results in Discussion (In line 264-272). The author needs to emphases the meaning of present study.

Response: In the first paragraph of the discussion, it is customary to summarize the results. We shortened this section to emphasize the main results and get to the heart of the discussion quicker.

In Line 296-311, the author mentioned that increased MLCs would play as a protective way and may arise inflammations. According to the results showed in Figure 2, the severe the NPDR is, the more MLCs detected. The author may need to re-organize this paragraph (line 296-311) clearer.

Response: We reorganized the discussion section to progress through the cell types based upon their role. We previously organized this section as monocyte-derived macrophages, microglia, perivascular macrophages, and hyalocytes because that is the order of their abundance during neuroinflammation. However, it makes more sense as this reviewer points out to organize this section based upon function. It now progress from monocyte-derived macrophages (pathogenic) to hyalocytes (pathogenic) to microglia (restorative) to perivascular macrophages (unknown). We thank the reviewer for their assistance in improving the discussion.

The author needs to add a Conclusion section in summarizing the key-findings or meaning of current study.

Response: A conclusion section has been added.

The author need to address clearer the value of current research in indicating the management of DM patients.

Response: MLCs are an emerging field and their identity is still hotly debated. We added a sentence to the conclusions reflecting that the role of MLCs in the pathogenesis of DME and DR are unknown and warrant further study based upon our correlative data.

Round 2

Reviewer 3 Report

Thank you for your carefully check and point-to-point response.